# Unseen No More: Unlocking the Potential of CLIP for Generative Zero-shot HOI Detection

## ABSTRACT

Zero-shot human-object interaction (HOI) detector is capable of generalizing to HOI categories even not encountered during training. Inspired by the impressive zero-shot capabilities offered by CLIP, latest methods strive to leverage CLIP embeddings for improving zero-shot HOI detection. However, these embedding-based methods train the classifier on seen classes only, inevitably resulting in seen-unseen confusion of the model during testing. Besides, we find that using prompt-tuning and adapters further increases the gap between seen and unseen accuracy. To tackle this challenge, we present the first generation-based model using CLIP for zero-shot HOI detection, coined HOIGen. It allows to unlock the potential of CLIP for feature generation instead of feature extraction only. To achieve it, we develop a CLIP-injected feature generator in accordance with the generation of human, object and union features. Then, we extract realistic features of seen samples and mix them with synthetic features together, allowing the model to train seen and unseen classes jointly. To enrich the HOI scores, we construct a generative prototype bank in a pairwise HOI recognition branch, and a multi-knowledge prototype bank in an image-wise HOI recognition branch, respectively. Extensive experiments on HICO-DET benchmark demonstrate our HOIGen achieves superior performance for both seen and unseen classes under various zero-shot settings, compared with other top-performing methods.

## CCS CONCEPTS

• **Computing methodologies** → **Computer vision**.

## KEYWORDS

Human-Object Interaction, Zero-shot Learning, Feature Generation

**ACM Reference Format:**
Anonymous Authors. 2024. Unseen No More: Unlocking the Potential of CLIP for Generative Zero-shot HOI Detection. In *Melbourne '24: ACM International Conference on Multimedia, 28 October - 1 November, 2024, Melbourne, Australia.* ACM, New York, NY, USA, 10 pages. https://doi.org/XXXXXXX.XXXXXXX

## 1 INTRODUCTION

Human-Object Interaction (HOI) detection, stemming from generic object detection, entails precisely localizing and categorizing humans and objects, and simultaneously inferring their relationships

**Unpublished working draft. Not for distribution.**

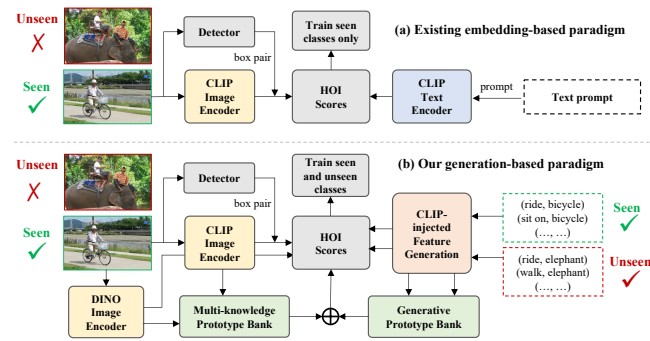

**Figure 1: Differences between existing embedding-based paradigm and our generation-based paradigm for zero-shot HOI detection. The former exploits CLIP to train visual and semantic embeddings of seen HOI categories only. Beyond that, our generation-based paradigm develops a new CLIP-injected feature generation module given either seen or unseen class names. The generated features enable the model to train seen and unseen HOI categories jointly. Besides, we construct generative prototype bank and multi-knowledge prototype bank to enrich the HOI scores.**

in images [6, 11, 12, 23, 24, 32, 41]. Existing studies on HOI detection are typically devoted to closed-domain scenarios, assuming all test classes have been seen at training stage. However, the methods are hardly applied to open-domain scenarios where some test classes are unseen and disjoint from seen ones. Considering the substantial time and effort involved in acquiring all classes in advance, zero-shot HOI detector remains a significant goal, as it is capable of generalizing to unseen HOI categories not encountered during training at all.

Thanks to its impressive generalization capability, contrastive language-image pre-training (CLIP) [38] has been a key remedy for various low-shot learning tasks [52, 53, 55]. Following this trend, a few works [27, 36, 37, 48] position CLIP as a valuable source of prior knowledge for zero-shot HOI detection, effectively discerning and understanding unseen interactions between human and objects. As depicted in Fig. 1(a), these methods follow an embedding-based paradigm, which utilizes CLIP embeddings to recognize seen HOI categories during training, and transfers the learned knowledge to unseen categories at inference. Despite recent advancements, this paradigm trains the classifier only on seen classes, inevitably leading the model to seen-unseen confusion during testing. That means some samples of unseen categories might be mis-classified into the set of seen categories. Moreover, some efforts suggest fine-tuning CLIP embeddings by incorporating learnable prompts or adding adapters, so as to further strengthen the adaptation on downstream tasks [13, 52, 53]. For instance, ADA-CM [27] integrates a learnable adapter into CLIP image encoder and achieves a significant

boost in the performance of both seen and seen HOI categories. However, we note that current methods overlook a key phenomenon: fine-tuning CLIP embeddings enlarges the performance gap between seen and unseen HOI categories. The primary reason is that, the fine-tuned embeddings become more influenced by image samples of seen categories, resulting in impaired generalization of the model to unseen categories. Hence, the remaining challenge for zero-shot HOI detection is *how we can adapt CLIP on seen categories and meanwhile retain its generalization on unseen ones.*

Instead of the embedding-based paradigm above, we reveal that leveraging generation-based paradigm for zero-shot HOI detection can alleviate the seen-unseen bias explicitly. To achieve it, we propose the first feature generation model for zero-shot HOI detection, coined HOIGen. As shown in Fig. 1(b), HOIGen aims to harness the potential of CLIP for generating image features, rather than utilizing CLIP for feature extraction solely. In this way, it trains the model for all HOI categories in a supervised learning manner. Specifically, we devise a CLIP-injected feature generator based on a variational autoencoder model, contributing the prior knowledge learned in CLIP to the feature generation of human-object pairs, humans, and objects. After the feature generator is trained, we use it to produce a variety of synthetic image features used by a HOI detection model, which comprises a pairwise HOI recognition branch and an image-wise HOI recognition branch. For the former branch, we integrate an off-the-shelf object detector (*i.e.* DETR) with a CLIP image encoder to obtain realistic image features of seen samples. Then we merge both synthetic and realistic features together and pass them into a generative prototype bank, resulting in several pairwise HOI scores. Subsequently, in the image-wise HOI recognition branch, we obtain two global image features from CLIP and DINO [5] image encoders, respectively, and thereby construct a multi-knowledge prototype bank, offering more comprehensive image-wise HOI scores. Eventually, we fuse all the HOI scores from the two branches to classify seen and unseen categories jointly.

Overall, this work has three-fold contributions below:

- This work is the first to address zero-shot HOI detection via a generation-based paradigm using CLIP. It alleviates the model overfitting on seen categories and improves the generalization on unseen ones.
- We devise a new CLIP-injected feature generator, synthesizing image features of humans, objects and their unions simultaneously. Besides, we utilize the synthetic features to construct a generative prototype bank for pairwise HOI recognition, and a multi-knowledge prototype bank for image-wise HOI recognition, respectively.
- Extensive experiments on HICO-DET benchmark exhibit superior performance of our HOIGen over prior state-of-the-arts. Our method achieves an absolute mAP gain of 2.54 on unseen categories and of 1.56 on seen categories.

## 2 RELATED WORK

### 2.1 HOI Detection and Beyond

HOI detection task is an integration of object detection, human-object pairing and interaction recognition. Broadly speaking, existing work can be categorized into one-stage and two-stage methods.

Initially, the two-stage methods [6, 11, 12, 27, 32, 36, 42, 45] predominated in HOI detection. Its first stage utilizes off-the-shelf object detectors to identify humans and objects according to their class labels. Subsequently, independent modules are designed in the second stage to classify interactions between each human-object pair. On the other hand, thanks to the recent emergence of DEtection TRansformer (DETR) [4], one-stage methods have witnessed a rapid evolution [7, 10, 22–24, 29, 41, 44, 54]. The methods are typically built upon a pre-trained DETR, predicting HOI triples directly in an end-to-end fashion.

**Zero-shot HOI.** Beyond the conventional setting above, an increasing line of work tends to research zero-shot HOI detection, allowing the model to infer some new HOI categories being not seen during training. To tackle this problem, early endeavors developed combinatorial learning, enabling the prediction of HOI triplets during inference [2, 20, 21, 35]. Inspired by the rapid advancements in visual-language pre-trained models like CLIP [38], more recent work has tapped into zero-shot generalization capabilities of these models to transfer prior knowledge for identifying unseen HOI categories [3, 27, 33, 36, 37, 42, 43, 48]. Specifically, these methods employ CLIP image encoder to encode raw images and subsequently extract features related to human-object pairs using bounding boxes acquired from an off-the-shelf detector. Subsequently, we achieve the HOI score with a given image and a candidate human-object interaction triplet. Although these embedding-based methods have shown promise of CLIP embeddings for zero-shot learning, they are inefficient for addressing the bias between seen and unseen classes. *Different from them, we devise a generative-based paradigm for zero-shot HOI detection, with an aim of exploiting CLIP for feature generation and alleviating the seen-unseen bias explicitly.*

### 2.2 Generative Zero-Shot Learning

Generative zero-shot learning allows the model to generate unseen samples from their corresponding attributes, converting the conventional zero-shot learning to a classic supervised learning problem. Generative adversarial networks (GANs) [16] and variational autoencoders (VAEs) [25] are two prominent members of generative models used for generalized zero-shot learning task which needs to infer both seen and unseen classes. Through this generative paradigm, the model can capture underlying data distributions and generate diverse samples resembling unseen categories [1, 9, 17, 28, 40, 49, 56]. In order to improve the generation quality and stability, a few methods combine the advantages of VAE and GAN for joint training to generate high-quality visual features for unseen categories [14, 15, 26]. Recently, some pioneering works study integrating CLIP to feature generation [19, 34, 39, 47]. For instance, CLIP-Forge [39] obtains the latent space of shapes by training an auto-encoder and then trains a normalized flow network to align the distribution of shape embeddings with image features from a pre-trained CLIP image encoder. Likewise, CLIP-GEN [47] obtains cross-modal embeddings through CLIP and converts the image into a sequence of discrete tokens in the VQGAN codebook space, training an autoregressive transformer and generating coherent image tokens. A recent work similar to ours is SHIP [46], which inserts pre-trained CLIP image-text encoders into a variational autoencoder (VAE) model, so as to generate synthetic image features

given arbitrary names of unseen classes. However, this generation method in SHIP is designed for simple image classification, but is challenged the complex compositions of human, objects and actions in HOI detection. *In this work, we develop a new CLIP-injected feature generation model which is tailored specifically for zero-shot HOI detection effectively.*

# 3 PRELIMINARIES

## 3.1 Problem Formulation

Given an image $I$ as input, its detection results are in the form of five-tuple $(b_h, s_h, b_o, s_o, c_o)$, where $b_h, b_o \in \mathbb{R}^4$ represent the bounding boxes of detected human and object instances; $c_o \in \{1, ..., O\}$ indicates the object category; $s_h$ and $s_o$ are the confidence scores of detected human and objects. The action category $c_a \in \{1, ..., A\}$ is then identified and a confidence score $s$ is used for each human-object pair. The final result is presented in the form of $HOI_{i,j} = < h, v_i, o_j >$ triplet, where $h$ represents the human, $v_i$ represents the category of the verb, and $o_j$ represents the category of the object. According to whether verbs and objects in the unseen categories $\mathbb{C}_{unseen}$ exist during training, zero-shot HOI detection can be further divided into three settings: (1) Unseen Composition (UC), where for all $(v_i, o_j) \in \mathbb{C}_{unseen}$, we have $v_i \in \mathbb{V}_{seen}$ and $o_j \in \mathbb{O}_{seen}$; (2) Unseen Object (UO), where for all $(v_i, o_j) \in \mathbb{C}_{unseen}$, we have $v_i \in \mathbb{V}_{seen}$ and $o_j \notin \mathbb{O}_{seen}$; (3) Unseen Verb (UV), where for all $(vi, oj) \in \mathbb{C}_{unseen}$, we have $v_i \notin \mathbb{V}_{seen}$ and $o_j \in \mathbb{O}_{seen}$. Note that, $\mathbb{C}_{unseen}$, $\mathbb{O}_{seen}$ and $\mathbb{V}_{seen}$ represent unseen combination categories, seen object categories, and seen verb categories, respectively.

## 3.2 Rethinking the Seen-unseen Bias

Generalized zero-shot learning suffers from inherently over-fitting on seen classes and weakly generalization on unseen ones. This problem becomes more difficult for HOI, as the number of HOI categories is much larger than that of object categories only. After a thorough examination of existing CLIP-based methods for zero-shot HOI detection [27, 36, 37, 42, 43], a consistent phenomenon we observe is they follow an embedding-based paradigm to infer unseen categories. Yet, these methods fail to avoid confusion between seen and unseen classes during testing, because the primary challenge persists in the absence of unseen categories during training. In addition, some of them leverage prompt tuning and adapters to better adapt CLIP on HOI benchmarks, resulting in an improved performance of both seen and unseen categories. Nevertheless, they overlook a significant consequence that the seen-unseen bias becomes more severe, because of adapting CLIP with seen data only. For instance, ADA-CM [27], which is a recent state-of-the-art, introduces two different settings. One is directly using CLIP in a training-free (TF) fashion, and the other is adding a learnable adapter to fine-tune (FT) the image embeddings obtained by CLIP. As shown in Fig. 2, thanks to the use of such an adapter, the seen accuracy obtains a substantial boost from 24.54% to 34.35%, and meanwhile the unseen accuracy increases from 26.83% to 27.63%. Nonetheless, we should be aware that the gap between seen and unseen accuracy also improves a lot, from -2.29% to 6.72%. Through such a relative comparison, we can conjecture that fine-tuning the CLIP embeddings benefits the adaptation ability with seen data,

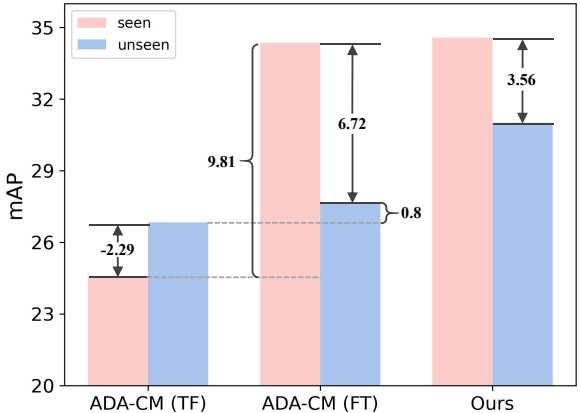

**Figure 2: Comparison of our method and ADA-CM on unseen and seen categories of HICO-DET dataset, under Non-rare First Unseen Combination (NF-UC) setting.**

whereas degenerates the generalization ability on unseen data. Differently, our method takes both adaptation and generalization into account. It can be seen that the unseen accuracy by our method has a more remarkable gain, making the seen-unseen gap decrease to 3.56%. We will elaborate on the details of our method below.

# 4 METHODOLOGY

## 4.1 Overview Framework

The framework of our proposed HOIGen, as depicted in Fig. 3, comprises three primary parts: CLIP-injected feature generation (Sec. 4.2), pairwise HOI recognition (Sec. 4.3) and image-wise HOI recognition (Sec. 4.4). Firstly, during the feature generation stage, we inject pre-trained CLIP into a variational autoencoder model, so as to produce synthetic features from CLIP text encoder and make them consistent with the real ones captured from CLIP image encoder. Secondly, for the pairwise HOI recognition, we adopt an off-the-shelf object detector (*i.e.* DETR) for bounding box detection. Then we employ CLIP image encoder for ROI-Align calculation and achieve real image features of seen samples, and collaborate the synthetic features to the real ones for training seen and unseen categories jointly. Besides, we use the CLIP-injected feature generator to construct a generative prototype bank, which aims to calculate the pairwise HOI recognition scores. Thirdly, the image-wise HOI recognition entails combining CLIP image encoder and DINO image encoder, and also involves the use of synthetic features. It develops a multi-knowledge prototype bank for computing the image-wise HOI recognition scores. Finally, we fuse the HOI scores of different branches for classifying the HOI category.

## 4.2 CLIP-injected Feature Generation

The idea of extracting CLIP feature embeddings for zero-shot HOI detection has been studies in previous work [27, 36, 37, 42]. These methods, however, fail to overcome the scarcity of unseen samples explicitly, resulting in a severe bias between seen and unseen HOI categories. Beyond that, our objective is to fully harness the potential of CLIP for feature generation, thereby leveraging prior

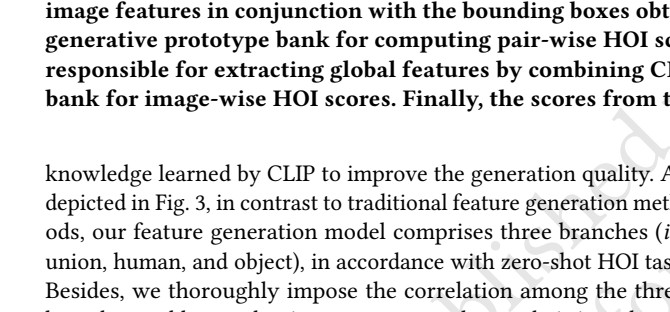

**Figure 3: Overview of the proposed HOIGen model, which primarily comprises CLIP-injected feature generation, pairwise HOI recognition branch and image-wise HOI recognition branch. We contribute CLIP image-text encoders to a variational autoencoder, which synthesizes image features in a two-stage fashion. The pairwise HOI recognition branch utilizes CLIP image features in conjunction with the bounding boxes obtained from a pre-trained DETR. The resulting features are fed into a generative prototype bank for computing pair-wise HOI scores. On the other hand, the image-wise HOI recognition branch is responsible for extracting global features by combining CLIP and DINO encoders, constructing a multi-knowledge prototype bank for image-wise HOI scores. Finally, the scores from the two branches are fused together to predict the HOI category.**

knowledge learned by CLIP to improve the generation quality. As depicted in Fig. 3, in contrast to traditional feature generation methods, our feature generation model comprises three branches (*i.e.* union, human, and object), in accordance with zero-shot HOI task. Besides, we thoroughly impose the correlation among the three branches and learn adaptive prompts to enhance their interdependence. Our feature generation is performed in two stages below.

**Stage I: Train VAE encoder $E(\cdot)$ and generator $G(\cdot)$.** In this stage, the feature generation process needs to deal with image regions of human-object union, human only, and object only, respectively, which are denoted as $img_u$, $img_h$ and $img_o$. For brevity, we present the regions with $img_k$ where $k \in (u, h, o)$. First of all, the CLIP image encoder $I(\cdot)$ extracts realistic image features $x_k = I(img_k)$ given $img_k$. Afterwards, we build a variational autoencoder model, where the encoder $E(\cdot)$ encodes $x_k$ into a latent code $z_k$, and the generator $G(z_k, c_k)$ then reconstructs $x_k$ from the latent code $z_k$ and the corresponding learnable class name $c_k$. Note that, different from previous work on generic image classification [46], our approach considers emphasizing meticulous consideration of "human" categories, where we observe distinct human expressions interacting with various objects. To achieve it, we categorize "human" prompts based on the corresponding object categories and introduce a novel prompt template (*e.g.* "person who interacts with <OBJECT>"). For example, if a person interacts with a bus, we label this person as "person who interacts with bus". This fine-grained categorization of humans facilitates more precise associations between humans and objects, and captures nuanced interactions between humans and specific objects. During

this stage, the optimization of both the encoder $E(x_k)$ and the generator $G(z_k, c_k)$ is achieved by minimizing the following evidence lower bound (ELBO) cost:

$$\mathcal{L}_{stageI} = \mathcal{L}_{KL} + \mathcal{L}_{recon}$$
$$= KL(E(x_k)||p(z_k|c_k)) + \mathbb{E}[-\log G(z_k, c_k)], \quad (1)$$

where $KL$ denotes the Kullback-Leibler divergence, $p(z_k|c_k)$ is a prior distribution that is assumed to be $N(0, 1)$, and $-\log G(z_k, c_k)$ estimates the reconstruction loss.

**Stage II: Freeze encoder $E(\cdot)$ and generator $G(\cdot)$, and fine-tune MLP.** This stage is dedicated to bridging the gap between synthetic and realistic images in HOIGen. It is noteworthy that the first stage aligns with features extracted by the CLIP image encoder, which may differ from the image features required in the following detector. To solve it, the same generator $G(\cdot)$ learned in Stage I remains frozen during Stage II training. The input transitions from a different dataset to a category prompt $c_k$ and a randomly initialized normal distribution $N(0, 1)$, and through the frozen generator $G(\cdot)$. Additionally, a new multilayer perceptron (MLP) is employed to align the reconstructed synthetic features with the realistic features. Throughout this process, only the MLP is optimized, while the other components remain frozen. This optimization process is realized through minimizing mean square error:

$$\mathcal{L}_{stageII} = \mathcal{L}_{MSE} = \frac{1}{n} \sum_{i=1}^{n} (\text{MLP}(x'_k) - \overline{x}_k)^2, \quad (2)$$

where $x'_k$ denotes the reconstructed features through generator $G(\cdot)$ and $\overline{x}_k$ indicates the realistic features extracted by CLIP in

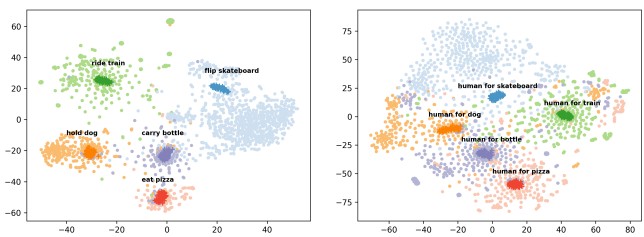

**Figure 4: Visualization of realistic (light regions) and synthesized features (dark regions) using *t*-SNE, with respect to unseen HOI categories from HICO-DET dataset [6]. We synthesize 100 features per category. The left shows the feature distributions of different pairs of <ACTION, OBJECT>, and the right presents the features of different objects.**

HOIGen. To show the quality of our generated features, we illustrate their distributions using *t*-SNE in Fig 4. We can see that, the feature distributions of different categories distinguish from each other clearly. In addition, synthesized features are distribution-aligned well with the corresponding realistic features in the space.

## 4.3 Pairwise HOI Recognition

Since most of existing methods are heavily affected by long-tail distributions, it becomes challenging to effectively manage complex scenes featuring a diverse range of interactive objects. To this end, we construct a **generative prototype bank** by using the generated features as key. It can alleviate the long-tail distribution, and avoid the time-consuming project of loading keys in the training samples. As shown in Fig 4, we generate 100 synthetic features for each HOI category through the CLIP-injected feature generator, and also generate the corresponding human and object synthetic features. We then extract the seen and unseen features from them as the key of the generative prototype bank. This design provides a powerful generative prototype bank that can better capture feature distributions. We convert the training labels into multi-hot encodings $L_{mh} \in \mathbb{R}^{N \times A}$ as the value for generative prototype bank. Inspired by previous research [27, 36], while extracting a rich visual feature prototypes, we also supplement semantic feature prototypes. Specifically, we first use handcrafted prompts (*i.e.* A photo of a person is <ACTION> an object) to generate the raw text description of interactions. Then we pass such prompt into CLIP text encoder $\mathcal{T}$ and obtain semantic prototype, thus enriching the prototype bank.

In training phase, given the feature map $\mathcal{M}$ extracted by the CLIP image encoder, and simultaneously utilizing DETR to extract the bounding boxes of the images. ROI-Align [18] is then applied to derive the union feature $v_u$, human feature $v_h$ and object feature $v_o$. These elements are collectively used in the computation of pairwise HOI recognition scores $s_P$:

$$s_P = \sum_i^M \lambda_i v_i P_i^T L_{mh} + \lambda_t v_u P_{Text}^T, M \in \{u, h, o\}, \quad (3)$$

where $\lambda_u, \lambda_h, \lambda_o$, and $\lambda_t$ adjust the weights of different terms; $P_{Text}$ signifies the utilization of manual prompts (*e.g.* A photo of a person is <ACTION> an object) to generate initial textual descriptions of interactions.

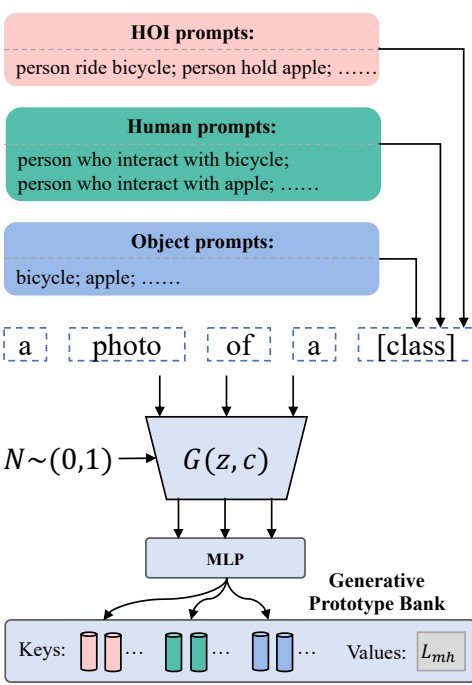

**Figure 5: Construction of generative prototype bank.**

## 4.4 Image-wise HOI Recognition

Global context information is a crucial aspect that cannot be overlooked in HOI detection. Hence, we not only utilize the global features provided by CLIP but also leverage the self-supervision capability of DINO [5] to enhance the contextual interactions in the images. Concretely, we construct a key-value **multi-knowledge prototype bank** for global knowledge ensemble. This bank contains the pre-learned knowledge from both CLIP and DINO because of caching two kinds of keys. Formally, we first utilize CLIP and DINO to independently extract visual features of training images, formulated as:

$$P_{DINO} = DINO(I_N), \quad (4)$$

$$P_{CLIP} = CLIP_{vis}(I_N), \quad (5)$$

where $I_N$ denotes training images, $CLIP_{vis}$ denotes the visual encoder of CLIP in HOIGen and $P_{CLIP}, P_{DINO} \in \mathbb{R}^{N \times A}$. In addition to the key, we utilize the same $L_{mh}$ used in generative prototype bank as the value for the multi-knowledge prototype bank.

In training phase, in order to fully extract global features, we use the image encoder of CLIP and DINO to generate feature vectors $v_{CLIP}, v_{DINO}$ for the input image $I_N$, respectively. Then use the multi-knowledge prototype bank $P_{CLIP}, P_{DINO}$ to perform image-wise HOI recognition. Finally, we obtain image-wise HOI scores $s_I \in R^N$ by

$$s_I = \sum_i^M \lambda_i v_i P_i^T L_{mh}, M \in \{CLIP, DINO\}, \quad (6)$$

where $\lambda_{CLIP}, \lambda_{DINO}$ balance the weights of different prototypes based on CLIP and DINO.

**Table 1: Zero-shot HOI detection results on HICO-DET benchmark. UC, UV and UO settings denote unseen composition, unseen verb, and unseen object, respectively. RF and NF represent rare first and non-rare first.**

(a) UC & UV & UO

| Method | Setting | Full | Seen | Unseen |
|---|---|---|---|---|
| ConsNet [35] | UC | 19.81 | 20.51 | 16.99 |
| EoID [48] | UC | 28.91 | 30.39 | 23.01 |
| HOICLIP [37] | UC | 29.93 | 31.65 | 23.15 |
| CLIP4HOI [36] | UC | 32.11 | 33.25 | 27.71 |
| **HOIGen (Ours)** | UC | **33.44** | **34.23** | **30.26** |
| GEN-VLKT [33] | UV | 28.74 | 30.23 | 20.96 |
| HOICLIP [37] | UV | 31.09 | 32.19 | 24.30 |
| CLIP4HOI [36] | UV | 30.42 | 31.14 | **26.02** |
| LOGICHOI [30] | UV | 30.77 | 31.88 | 24.57 |
| **HOIGen (Ours)** | UV | **32.34** | **34.31** | 20.27 |
| CLIP4HOI [36] | UO | 32.58 | 32.73 | 31.79 |
| **HOIGen (Ours)** | UO | **33.48** | **32.90** | **36.35** |

(b) RF-UC & NF-UC

| Method | Setting | Full | Seen | Unseen |
|---|---|---|---|---|
| GEN-VLKT [33] | RF-UC | 30.56 | 32.91 | 21.36 |
| HOICLIP [37] | RF-UC | 32.99 | 34.85 | 25.53 |
| ADA-CM [27] | RF-UC | 33.01 | 34.35 | 27.63 |
| CLIP4HOI [36] | RF-UC | **34.08** | **35.48** | 28.47 |
| LOGICHOI [30] | RF-UC | 33.17 | 34.93 | 25.97 |
| **HOIGen (Ours)** | RF-UC | 33.86 | 34.57 | **31.01** |
| GEN-VLKT [33] | NF-UC | 23.71 | 23.38 | 25.05 |
| HOICLIP [37] | NF-UC | 27.75 | 28.10 | 26.39 |
| ADA-CM [27] | NF-UC | 31.39 | 31.13 | 32.41 |
| CLIP4HOI [36] | NF-UC | 28.90 | 28.26 | 31.44 |
| LOGICHOI [30] | NF-UC | 27.95 | 27.86 | 26.84 |
| **HOIGen (Ours)** | NF-UC | **33.08** | **32.86** | **33.98** |

## 4.5 Training Objective

To classify the HOI categories, we need to combine the HOI scores from the pairwise and image-wise branches together. The whole objective is to optimize the network parameters $\theta$ via the cross-entropy loss $\mathcal{L}_{total}$ below:

$$\theta^* = \arg\min_{\theta} \mathcal{L}_{total}(s(\bigcup_{j}^{M} F_j(I), K), L_{GT}), M \in \{I, P, D\}, \quad (7)$$

where $I$ represents the input image, $F_I, F_P, F_D$ respectively represent CLIP local feature extraction, CLIP global feature extraction and DINO feature extraction, $K$ represents the prototype bank. $s$ is the score combining pairwise and image-wise HOI scores, where $s = s_P + s_I$. $L_{GT}$ represents the ground truth label.

## 5 EXPERIMENTS

### 5.1 Experimental Protocol

**Datasets and Metric.** We conduct extensive experiments on the most benchmarked and challenging dataset, HICO-DET [6]. It is composed of 47,776 images, with 38,118 designated for training and 9,658 for testing. The annotations for HICO-DET encompass 600 categories of HOI triplets, derived from 80 object categories and 117 action categories. Besides, 138 out of the 600 HOI categories are classified as Rare due to having fewer than 10 training instances, while the remaining 462 categories are labeled as Non-Rare. Typically, the methods are evaluated with mean Average Precision (mAP) metric.

**Zero-shot Settings.** Following the protocols established in previous work [30, 36, 37], To make a fair and comprehensive comparison with previous work, we construct zero-shot HOI detection with five manners: Rare First Unseen Combination (RF-UC), Non-rare First Unseen Combination (NF-UC), Unseen Combination (UC), Unseen Verb (UV), and Unseen Object (UO). Specifically, UC indicates all action and object categories are included during training, but some HOI triplets (*i.e.* combinations) are absent; Under the RF-UC setting, the tail HOI categories are selected as unseen categories, while NF-UC uses head HOI categories being unseen. The UV (or

UO) setting indicate that some action (or object) categories are not concluded in the training set.

**Implementation Details.** We employ DETR with ResNet-50 as backbone, and the CLIP variant with an image encoder based on ViT-B/16. The DINO model we use is also pre-trained with ResNet-50 backbone. First of all, we need to train the CLIP-injected VAE model separately. We optimize it via the AdamW optimizer with a learning rate of $10^{-3}$ for 50 epochs. The batch size is set to 256. The dimension of the latent code $z$ is equal to that of the token embeddings. Afterwards, we generate 100 features for each of both seen and unseen HOI categories, and randomly choose some of them to merge with some realistic features. Remarkably, we **do not use any external data or model** since the generated model is also trained on the same training dataset. We then optimize the whole HOI detection model for 15 training epochs through AdamW with an initial learning rate of $10^{-3}$. This batch size in this training stage is 4. All experiments are conducted on a single NVIDIA RTX 4090 GPU card. Following [27], $\lambda_{CLIP}, \lambda_{DINO}, \lambda_u, \lambda_h, \lambda_o$ are all set to 0.5, while $\lambda_t$ is set to be 1.0, for a fair comparison.

### 5.2 Comparison with the State-of-the-arts

**Zero-shot HOI Detection.** We show a comprehensive comparison in Table 1, where our proposed HOIGen is superior to or on par with the competitors under five zero-shot settings. First, in terms of UC, RF-UC and NF-UC settings, our approach marks a significant advancement, particularly for the performance of unseen categories. Relative to the state-of-the-art methods, CLIP4HOI [36] and ADA-CM [27], our unseen accuracy achieves a consistent gain of 2.55%, 2.54%, 1.57% for UC, RF-UC, and NF-UC, respectively. We note that our results are also very competitive performance in both the seen categories and overall metrics. Moreover, our approach benefits improving the mAP accuracy of Full and Seen, as we also generate additional features for seen categories apart from unseen ones. Second, considering the UV setting, our relatively inferior performance in the unseen accuracy may be attributed to the abstract nature of actions, devoid of specific characteristics like objects. Besides, the same action can be applied to various objects, posing a significant

**Table 2: Fully-supervised HOI detection results on the HICO-DET dataset. All the methods are built with ResNet-50.**

| Method | Backbone | Full | Rare | Non-rare |
|--------|----------|------|------|----------|
| IDN [31] | ResNet-50 | 23.36 | 22.47 | 23.63 |
| HOTR [23] | ResNet-50 | 25.10 | 17.34 | 27.42 |
| ATL [20] | ResNet-50 | 28.53 | 21.64 | 30.59 |
| AS-Net [8] | ResNet-50 | 28.87 | 24.25 | 30.25 |
| QPIC [41] | ResNet-50 | 29.07 | 21.85 | 31.23 |
| MSTR [24] | ResNet-50 | 31.17 | 25.31 | 32.92 |
| UPT [50] | ResNet-50 | 31.66 | 25.94 | 33.36 |
| PVIC [51] | ResNet-50 | 34.69 | 32.14 | 35.45 |
| GEN-VLKT [33] | ResNet-50 | 33.75 | 29.25 | 35.10 |
| ADA-CM [27] | ResNet-50 | 33.80 | 31.72 | 34.42 |
| HOICLIP [37] | ResNet-50 | 34.69 | 31.12 | _35.74_ |
| CLIP4HOI [36] | ResNet-50 | _35.33_ | _33.95_ | _35.74_ |
| LOGICHOI [30] | ResNet-50 | **35.47** | 32.03 | **36.22** |
| **HOIGen (Ours)** | ResNet-50 | 34.84 | **34.52** | 34.94 |

challenge in recognizing unseen verbs. Nonetheless, our HOIGen has achieved new state-of-the-art results on Full and Seen mAP, competing against HOICLIP [37]. Third, we conduct a comparison with CLIP4HOI [36] under the UO setting. It reveals that our approach obtains superior results for all the three mAP metrics, particularly evident for the unseen results, achieving a significant increase of 4.56%.

**Fully-supervised HOI Detection.** Apart from zero-shot settings, our feature generation can be also applicable in a fully-supervised setting. We present the comparative results in Table 2. Overall, LOGICHOI [30]achieves the highest Full accuracy among other methods because it is designed specifically for normal HOI detection. However, this method fails to retain the superior performance under zero-shot setting (see Table 1). Notably, our HOIGen becomes the new state-of-the-art for rare categories, improved by approximately 0.6%. This suggests that the generated features effectively complement the limited number of real samples of rare categories, even within this fully-supervised setting.

## 5.3 Ablation Study

We undertake a series of ablation experiments to verify the effectiveness of the HOIGen model. All experiments are conducted under the context of the NF-UC setting.

**Component analysis.** First of all, we conduct experiments to delineate the contribution of each component to the model. As reported in Table 3, we study the impact of feature generation, CLIP-based and DINO-based image feature individually. It can be seen that the feature generation module acts as the most influential factor in raising the unseen accuracy performance, resulting in a 0.9% improvement over the baseline (*i.e.* the first row in the table). This finding underscores our motivation of exploring feature generation for zero-shot HOI detection. In addition, the use of CLIP-based image feature is observed to augment the seen accuracy by 0.61%. Moreover, by integrating DINO-based image feature, we further obtain 0.8%-0.9% gain in seen accuracy. These results reveal the

**Table 3: Component analysis under the NF-UC setting on the HICO-DET dataset. FG represents feature generation module, and CLIP-img and DINO-img mean extracting image feature from pre-trained CLIP and DINO image encoder, respectively.**

| Exp | FG | CLIP-Img | DINO-Img | Full | Seen | Unseen |
|-----|-----|----------|----------|------|------|--------|
| 1 | | | | 31.39 | 31.13 | 32.41 |
| 2 | √ | | | 31.77 | 31.39 | 33.31 |
| 3 | | √ | | 32.00 | 31.88 | 32.45 |
| 4 | | | √ | 31.97 | 31.98 | 31.92 |
| 5 | √ | | √ | 31.67 | 31.28 | 33.21 |
| 6 | | √ | √ | 32.02 | _32.00_ | 32.09 |
| 7 | √ | √ | | _32.26_ | 31.95 | _33.50_ |
| 8 | √ | √ | √ | **33.08** | **32.86** | **33.98** |

benefit of fusing more prior knowledge learned from different large-scale pre-trained models.

**Number of generated features.** As shown in Table 4a, we investigate the influence of varying the number of generated features on the HOIGen model. Specifically, we generate $N_{bs}$ features for each sample in the batch. In this way, the total of samples in each training batch becomes $(N_{bs} + 1) \times batch - size$. We see that, the unseen accuracy yields improvement along with the addition of generated features, suggesting a positive impact on unseen categories. However, as more generated features are integrated, the Full and Seen performance gradually declines. The main reason is that the feature distributions of real data becomes disrupted, when more and more generated features dominate the feature space. Finally, we choose $N_{bs} = 1$ in the experiments.

**Construction of prototype bank.** This experiment is to validate the construction of our generative prototype bank. In Table 4b, we compare several construction methods, including using realistic features alone ('R'), adding realistic and generated features ('R+G'), concatenating realistic and generated features ('R ⊕ G'), and using generated features alone ('G'). Experimental results demonstrate that after adding generated features, the performance of the network increases significantly. This may be because the generated features contain unseen sample features, making the generative prototype bank richer. As for the reason why using only generated features as a prototype bank performs best, it can be considered that the fusion of both realistic and generated features may lead to the confusion in the feature distribution.

**Size of generative prototype bank.** This experiment aims to study the impact of the size $N_{size}$ of generative prototype bank. As shown in Table 4c, the highest scores occurs when $N_{size} = 2$. When we expand the bank size further (*i.e.* 3 and 4), the performance witnesses a decline on all the three metrics. The main reason might be the noisy prototypes involved in the bank. Fortunately, we can control the size to avoid this problem effectively.

## 5.4 Qualitative Results

In addition to quantitative results above, we further elaborate some qualitative results as shown in Fig. 5. We observe the following: (1) Our method demonstrates effectiveness in identifying

**Table 4: Ablative experiments for (a) number of generated features, (b) different prototype bank construction, and (c) prototype bank size. $N_{bs}$ represents the number of synthetic features for each sample in the batch, $N_{size}$ indicates the number of samples in the prototype bank per HOI. 'R' and 'G' is short for realistic and generated features, respectively . All the results are evaluated on HICO-DET under the NF-UC setting.**

<table>
<tr><td colspan="4" align="center">(a) Number of generated features.</td><td colspan="4" align="center">(b) Different prototype bank construction.</td><td colspan="4" align="center">(c) Prototype bank size.</td></tr>
<tr><td>$N_{bs}$</td><td>Full</td><td>Seen</td><td>Unseen</td><td>Construction</td><td>Full</td><td>Seen</td><td>Unseen</td><td>$N_{size}$</td><td>Full</td><td>Seen</td><td>Unseen</td></tr>
<tr><td>1</td><td>**33.08**</td><td>**32.86**</td><td>33.98</td><td>$R$</td><td>32.28</td><td>32.02</td><td>33.33</td><td>1</td><td>32.25</td><td>31.97</td><td>33.37</td></tr>
<tr><td>2</td><td>32.56</td><td>32.19</td><td>34.06</td><td>$R + G$</td><td>32.39</td><td>32.07</td><td>33.66</td><td>2</td><td>**33.08**</td><td>**32.86**</td><td>**33.98**</td></tr>
<tr><td>3</td><td>31.77</td><td>31.17</td><td>**34.16**</td><td>$R \oplus G$</td><td>32.59</td><td>32.41</td><td>33.30</td><td>3</td><td>32.07</td><td>31.60</td><td>33.95</td></tr>
<tr><td>4</td><td>31.54</td><td>30.99</td><td>33.71</td><td>$G$</td><td>**33.08**</td><td>**32.86**</td><td>**33.98**</td><td>4</td><td>32.26</td><td>31.95</td><td>33.50</td></tr>
</table>

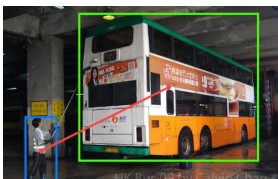 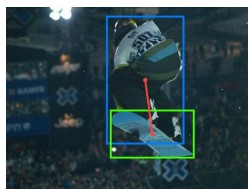 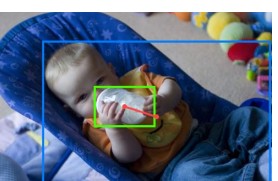 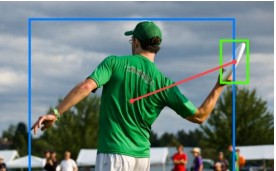

Ground Truth:
<wash,bus>
Prediction:
<wash,bus>

Ground Truth:
<stand on/ride/jump/hold/wear,snowboard>
Prediction:
<stand on/ride/jump/hold/wear,snowboard>

Ground Truth:
<hold/drink with,bottle>
Prediction:
<hold/drink with,bottle>

Ground Truth:
<hold/throw,frisbee>
Prediction:
<hold/spin,frisbee>

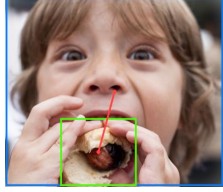 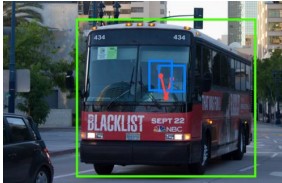 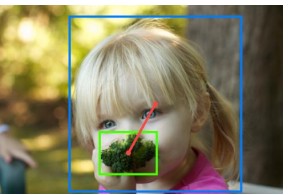 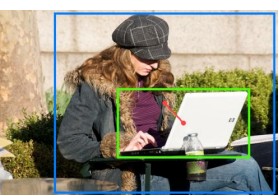

Ground Truth:
<carry/eat/hold,hot dog>
Prediction:
<carry/eat/hold,hot dog>

Ground Truth:
<sit on/drive,bus>
Prediction:
<sit on/drive,bus>

Ground Truth:
<smell/eat/hold,broccoli>
Prediction:
<smell/eat/hold,broccoli>

Ground Truth:
<read/type on,laptop>
Prediction:
<read/type on/open,laptop>

**Figure 6: Visualization of detection results. Correctly classified seen and unseen category are marked in blue and green respectively, rare category classification results are marked in orange, and incorrect recognition results are marked in red. Images sampled from HICO-DET dataset and visualized under NF-UC setting.**

samples of rare categories within the dataset. For example <person,wash,bus> and <person,smell,broccoli>, they are rare HOI combinations, especially <person, smell,broccoli> is easily confused with <person,eat,broccoli>. (2) Even in the multi-label scenario, our method exhibits notable discriminative capability towards unseen categories. In particular for the second image in the first row containing five HOIs , four of which belong to the unseen categories, our method recognizes them accurately. (3) Some semantically similar actions may lead to recognition errors, such as <person, throw, frisbee> and <person, spin, frisbee>. However, other semantically similar actions can still be accurately recognized, such as <person, hold, hot dog> and <person, carry, hot dog>. The majority of these human-object pairs are accurately recognized by our method.

## 6 CONCLUSION

We have proposed HOIGen, a pioneering method for zero-shot HOI detection that harnesses the capabilities of CLIP for feature generation. By fully leveraging the generated synthetic features, our approach effectively captures the feature distributions in the dataset, facilitating robust performance in zero-shot HOI detection scenarios across both seen and unseen categories of HOI. The generated features are fed into a generative prototype bank for computing pairwise HOI scores. Remarkably, we do not use any external data or model since the generated model is also trained on the same training dataset. Besides, the image-wise HOI recognition branch is responsible for extracting global features by combining CLIP and DINO encoders, thereby constructing a multi-knowledge prototype bank for image-wise HOI scores. Finally, the scores from the two branches are fused together to predict the HOI category. Extensive experiments conducted on the HICO-DET benchmark validate the effectiveness of HOIGen. In future work, it is promising to generate more diversified visual features aligning better with the distributions of realistic features.

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
