# OpenReview forum: "Unseen No More: Unlocking the Potential of CLIP for Generative Zero-shot HOI Detection"
_acmmm.org/ACMMM/2024/Conference — MM2024 Poster_

### Official Review · Reviewer_Bhfj · 2024-05-22

**Rating:** 4
**Confidence:** 3

**Summary:**

This paper presents a generation-based model using CLIP for zero-shot HOI detection It uses CLIP for feature generation instead of feature extraction only. To achieve it, a CLIP-injected feature generator is proposed in accordance with the generation of human, object and union features. Then,   realistic features of seen samples  are extracted and mixed with synthetic features together, allowing the model to train
seen and unseen classes jointly. To enrich the HOI scores, a generative prototype bank is constructed in a pairwise HOI recognition branch, and a multi-knowledge prototype bank in an image-wise HOI recognition branch, respectively. Experiments show the effectiveness of the proposed method.

**Strengths:**

1. The paper is overall well-written and easy to follow.

2. The experiments are comprehensive and demonstrated the effectiveness of the proposed method.

**Limitations:**

1. The reviewer concerns about the scope of the paper. As the paper is submitting to ACM MM, but there is barely literature connection to previously ACM MM papers. To better align with the scope of ACM MM, the reviewer recommends adding more MM literatures for comparison and discussion to align with the scope of ACM MM.

2. In Fig.2, the authors claim that by taking both adaption and generalization into account, the performance is increased compared with CLIP-based method. However, the reviewer notice that the proposed method also adopts a few off-the-shelf models which incorporate external training information into the task. Thus, the authors need to clarify how the adaption and generalization is enhanced, rather than simply relying on external knowledge leaked from other large models.

3. The reviewer noticed that for unseen verb, the unseen performance of the proposed method underperforms all the baseline methods. Particularly, the unseen performance is around 6% lower than the best method CLIP4HOI. Why is this phenomenon? Is the method good at detecting nouns but bad at verbs?


Typos: Line342 "The idea of extracting CLIP feature embeddings for zero-shot HOI
detection has been studies in previous work" -> "studied."

**Suitability:**

2

---

### Official Review · Reviewer_mmny · 2024-05-26

**Rating:** 4
**Confidence:** 3

**Summary:**

This paper propose a CLIP-injected feature generator to train models on seen and unseen features which improves models’ generalizability. Also, the paper develops two banks, generative prototype bank and multi-knowledge prototype bank. The former solves pairwise HOI recognition and the later focus on image-wise HOI recognition.

**Strengths:**

1 Clear writing and good representation:
Both figures and tables are well designed and the captions are easy to read and help to understand the main ideas.

2 Good performance:
In Table 1, this paper achieves significant improvement on the unseen of UC and UO settings which show the good generalizability of HOIGen. Additionally, on RF-UC and NF-UC, HOIGen also show the good generalizable performance with 2.54% and 1.57% improvements compared with previous methods.

3 Novel idea:
The paper proposes two banks--generative prototype bank and multi-knowledge prototype bank, to solve pairwise HOI recognition and image-wise HOI recognition, which is interesting to me.

**Limitations:**

1, The discussion is missing about the relatively low performance on UV unseen. Can authors show more detail discussion about the performance degradation in UV settings, such as the specific examples?

**Suitability:**

2

---

### Official Review · Reviewer_iYXb · 2024-05-29

**Rating:** 3
**Confidence:** 3

**Summary:**

The paper present HOIGen, the first generation-based model using CLIP for zero-shot HOI detection. Unlike traditional methods that only use CLIP for feature extraction, HOIGen leverages CLIP for feature generation. By mixing realistic features of seen samples with synthetic features, the model can train on both seen and unseen classes jointly. To enhance HOI scores, the paper constructs a generative prototype bank for pairwise HOI recognition and a multi-knowledge prototype bank for image-wise HOI recognition. Extensive experiments on the HICO-DET benchmark demonstrate that HOIGen achieves superior performance.

**Strengths:**

1.	The paper first proposes a novel generation-based paradigm in HOI detection.
2.	The experimental results are impressive.
3.	Ablation study verifies the effectiveness of each module.

**Limitations:**

1. When training the VAE for human categories, why choose the prompt template "person who interacts with <OBJECT>" instead of "person who <VERB> an object"?
2. Is the generator implemented as the CLIP text encoder? If so, how does it generate $ x_{k}'$?
3. Which dataset is used to train stage 2 of "CLIP-injected Feature Generation"?
4. The paper claims that "the features of stage 1 may differ from the image features required in the following detector." How does stage 2 training address this problem?
5. After loading the generated features into the prototype bank, how is supervision provided for the unseen classes during training?
6. The experiment is only conducted on a single dataset. What is the model's performance on V-COCO?

**Suitability:**

2

---

### Meta-Review · Area_Chair_dEs6 · 2024-07-02

**Recommendation:** Accept (Poster)
**Confidence:** 5

**Metareview:**

The initial ratings for this paper were ba, ba, br. After reading the authors' feedback, all reviewers learned to accept this paper. AC is therefore happy to recommend acceptance and invites the authors to incorporate the materials from the rebuttal to the camera ready.